# Spatial Heterogeneity of b Values in Northeastern Tibetan Plateau and Its Interpretation

**DOI:** 10.3390/e26030182

**Published:** 2024-02-21

**Authors:** Nan Hu, Peng Han, Rui Wang, Fuqiang Shi, Lichun Chen, Hongyi Li

**Affiliations:** 1Key Laboratory of Intraplate Volcanoes and Earthquakes, Ministry of Education, China University of Geosciences (Beijing), Beijing 100083, China; hn@email.cugb.edu.cn (N.H.); lih@cugb.edu.cn (H.L.); 2Shaanxi Earthquake Agency, Xi’an 710068, China; shifuqiang121@163.com; 3School of Geophysics and Information Technology, China University of Geosciences (Beijing), Beijing 100083, China; 4Department of Earth and Space Science, Southern University of Science and Technology, Shenzhen 518055, China; 11930870@mail.sustech.edu.cn; 5College of Earth Sciences, Guilin University of Technology, Guilin 541004, China; glutclc@glut.edu.cn

**Keywords:** *b* value, HIST-PPM, Zmap, the Northeastern Tibetan Plateau, fault zone

## Abstract

The northeastern margin of the Tibetan Plateau (NE Tibetan Plateau) exhibits active geological structures and has experienced multiple strong earthquakes, with *M* ≥ 7, throughout history. Particularly noteworthy is the 1920 *M*8^1^/_2_ earthquake in the Haiyuan region that occurred a century ago and is documented as one of the deadliest earthquakes. Consequently, analyzing seismic risks in the northeastern margin of the Tibetan Plateau holds significant importance. The *b* value, a crucial parameter for seismic activity, plays a pivotal role in seismic hazard analyses. This study calculates the spatial *b* values in this region based on earthquake catalogs since 1970. The study area encompasses several major active faults, and due to variations in *b* values across different fault types, traditional grid-search methods may introduce significant errors in calculating the spatial *b* value within complex fault systems. To address this, we employed the hierarchical space–time point–process (HIST-PPM) method proposed by Ogata. This method avoids partitioning earthquake samples, optimizes parameters using Akaike’s Bayesian Information Criterion (ABIC) with entropy maximization, and theoretically allows for a higher spatial resolution and more accurate *b* value calculations. The results indicate a high spatial heterogeneity in *b* values within the study area. The northwestern and southeastern regions exhibit higher *b* values. Along the Haiyuan fault zone, the central rupture zone of the Haiyuan earthquake has relatively higher *b* values than other regions of this fault zone, which is possibly related to the sufficient release of stress during the main rupture of the Haiyuan earthquake. The *b* values vary from high in the west to low in the east along the Zhongwei fault. On the West Qinling fault zone, the epicenter of the recent Minxian–Zhangxian earthquake is associated with a low *b* value. In general, regions with low *b* values correspond well to areas with moderate–strong seismic events in the past 50 years. The spatial differences in *b* values may reflect variances in seismic hazards among fault zones and regions within the same fault zone.

## 1. Introduction

It is widely accepted that the magnitude–frequency distribution of crustal earthquakes obeys an exponential law size relation, which is described in terms of magnitude by log*N* = *a* − *bM*, where *N* is the number of earthquakes greater or equal to magnitude *M* and *a* and *b* are the constants [1,2]. Through laboratory experiments and seismic research, it has been demonstrated that the *b* value is inversely linked with underground differential stress levels [3,4,5,6,7]. A highly stressed area where eventual ruptures (large earthquakes) are often observed to nucleate is characterized by a low *b* value of the earthquake frequency–size distribution [8,9,10]. Therefore, a declining *b* value or a low *b* value suggests an increased earthquake risk. The *b* value is also considered as an indicator of the conditions in the crust, which are directly or indirectly related to the stress state, such as the faulting style [11,12], locked or creeping fault patches [13,14,15,16], material properties [7,17], and pore–pressure perturbations [18,19,20,21], among others [22]. Therefore, the *b* value is essential in improving our physical understanding of earthquake occurrences and is often utilized for medium- to long-term earthquake predictions [8,9,23].

The *b* value is the rate parameter of an exponential distribution. An estimation of the *b* value is affected by several factors, including data quality [14,24], the sample size [24,25,26,27], approaches of the estimation (such as the maximum likelihood estimator and the least square method), and the completeness of magnitude. These factors highlight that *b* value analyses highly depend on some subjective choices and may vary obviously from one expert to another in the same region [28]. Conventional methods to estimate the spatial *b* value are fixed grid searches, fixing the number of earthquakes or the radius to sample the seismic event to estimate the *b* value at the grid points or adaptive window and changing its size and shape to take into account the differences in the statistical estimates of *b* values in adjacent grid nodes [29,30].

To obtain a robust *b* value spatial map, a completeness of the magnitude of the earthquake sample is critical [10,31]. Therefore, a fixed grid search for individual grids should consider the adequacy of seismic events and the grid size to ensure spatial resolution. Different fixed numbers or search radii may result in unstable *b* values, creating ambiguity. In practice, to obtain a higher spatial resolution, an overlap of the seismic events used to calculate the *b* value of adjacent grid points cannot be neglected [32]. It is reported that the *b* value varies systematically for different faulting styles [12], which may cause a bias in the *b* value because of the individual grid sample allotted in different style faults or segments, especially in complex-fault-system regions [33].

For estimating and interpolating *b* values in space, Ogata presented the hierarchical space–time point–process model (HIST-PPM), cubic B-spline expansions, and the Bayesian technique [34,35,36,37,38]. Such modeling is suited for observing highly clustered points with accurate locations. It may be more objective and precise to compute spatial *b* values, since it uses Delaunay tessellation to interpolate the *b* value at the nearest three earthquakes and does not call for the gridding of seismic events [39], especially for highly fractured regions with different types of active faults. We implemented the HIST-PPM method to evaluate the *b* value based on more than fifty years of catalogs and studied whether the low *b* value areas have correlations with moderate–strong earthquakes in the NE Tibetan Plateau.

The NE Tibetan Plateau is of elevated topography and great crustal thickness [40], resulting from the ongoing India–Eurasia collision, and is bordered by the rigid Alxa and Ordos blocks in the north and northeast, respectively [40,41]. Influenced by the continuous northward thrusting of the Indian plate, a series of tectonic zones has developed between the Qilian Mountains in the NE Tibetan Plateau and the Alxa block in the southwest part of the North China Craton, such as fault belts, folds, and Cenozoic sedimentary basins of different scales and natures [42,43,44,45,46,47]. One dramatic phenomenon is the several large NW-trending strike-slip and thrust faults (e.g., the extensive Haiyuan fault and the Western Qinling; Figure 1) situated in this region [43]. These fault belts extend for hundreds of kilometers with different style segments, such as the extensive Haiyuan fault. The extensive Haiyuan fault includes the Maomao Mountain fault and JinQiang River fault to the west, exhibiting lateral thrust striking [40,46], the Laohu Mountain fault and Haiyuan fault in the middle with lateral striking [48], and the Liupanshan fault to the east with lateral thrust striking [49,50]. HIST-PPM is suitable to estimate the *b* value for this region, since the NE Tibetan Plateau is tectonically complex and is attributed to different stress regimes. Since inconsistencies in the *b* value are often observed, and to also ensure reliability and to contrast with other methods, Zmap was also used to estimate the *b* value.

**Figure 1 entropy-26-00182-f001:**
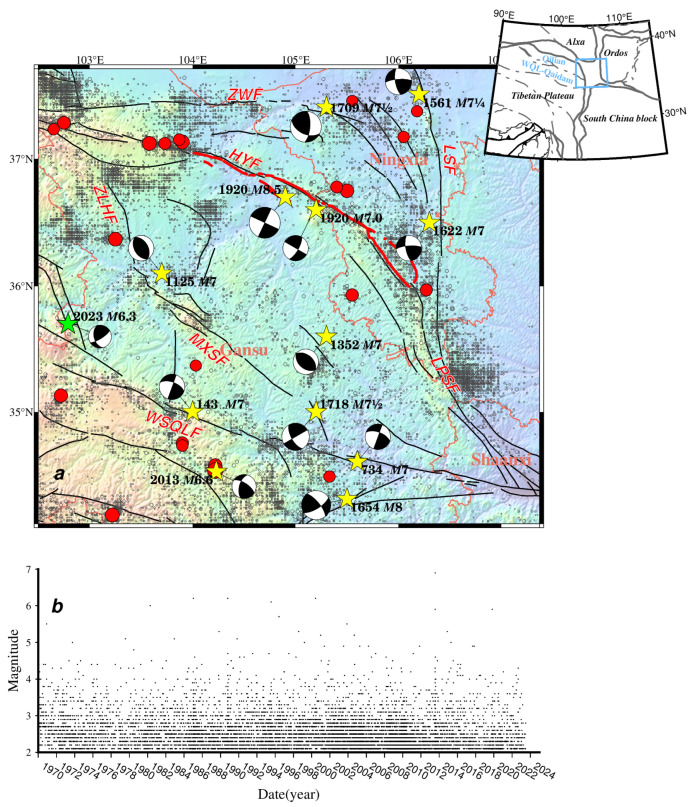
The spatial and temporal distribution of earthquakes in the NE Tibetan Plateau from January 1970 to June 2023 with main faults and focal mechanisms of large historical earthquakes. In the upper map (**a**), fault belts discussed in this paper are drawn in dark gray traces with abbreviated names, modified from Deng [51]; small hollow circles denote earthquakes with a magnitude of less than 5.0; red solid circles denote earthquakes with a magnitude equal to or larger than 5.0 from January 1970 to June 2023; the green solid star indicates the 18 December 2023 *M*6.3 earthquake that happened in the Linxia Hui autonomous region in Gansu, which was not included for *b* value estimation, and the size of the circle was scaled to the magnitude; yellow stars represent historic large earthquakes based on the Catalogue of Chinese Historical Strong Earthquakes from the 23^RD^ century BC to 1911 and the Catalogue of Chinese Present Strong Earthquakes from 1912 to 1990 [52,53]; focal-mechanism solutions of large earthquakes are shown as beach balls, estimated by empirical formulae using geometric information of seismic faults, tectonic stress, and rupture scales, referenced from previous studies, which are listed in Table 1; bold red lines denote the 230 km rupture zone of the 1920 *M*8^1^/_2_ Haiyuan earthquake [54,55,56]; and provincial borders are shown by orange lines with names in the center of each province. The inset figure shows active blocks in the NE Tibetan Plateau and the blue frame circles the extent of the study area. The lower map (**b**) is the temporal–magnitude distribution of seismic events. Abbreviated names of faults are labeled as follows: ZWF = Zhongwei fault; HYF = Haiyuan fault; LSF = Luoshan fault; LPSF = Liupanshan fault; ZLHF = Zhuanglanghe fault; MXSF = Maxianshan fault; WQLF = Western Qinling fault.

According to historical records, the NE Tibetan Plateau has experienced several significant large earthquakes and is regarded as one of China’s most seismically risky areas. Twelve large earthquakes above *M*7.0 are plotted in Figure 1, with focal mechanisms shown (Table 1). The focal mechanisms vary by faults. The 1709 Zhongwei *M*7^1^/_2_ earthquake was a thrust-strike slip on the ZWF at the northernmost region of the NE Tibetan Plateau. One of the deadliest earthquakes was the 1920 *M*8^1^/_2_ Haiyuan earthquake in Gansu province, which killed approximately 230,000 people [57]. More than 76 global stations recorded this large earthquake [58]. This was a left-lateral strike-slip earthquake that happened on the Haiyuan fault. The 1622 Guyuan *M*7.0 earthquake was also a left-lateral strike-slip with a thrust component at the Luoshan fault. In the Longxi region, lying in the region’s center, two historic earthquakes above *M*7.0 were of a strike-slip thrust. In the eastern part, around the Western Qingling fault, focal mechanisms of five historic earthquakes above *M*7.0 were from strike to thrust. The variety of focal mechanisms suggests a complex stress environment in this region.

**Table 1 entropy-26-00182-t001:** Focal-mechanism solutions of historical earthquakes and two notable earthquakes are plotted in Figure 1.

Time (Year-Month-Day)	Epicenter (Longitude°, Latitude°)	Strike	Dip	Slip	Magnitude	City	References
143-10-01 ^1^	104.0, 35.0	40	65	38.6	7	Weiyuan	[59,60,61]
734-03-23 ^1^	105.5, 34.5	40	65	30	7	Tianshui	[59,60,62]
1125-09-06 ^1^	103.6, 36.1	295	80	15	7	Lanzhou	[63,64,65]
1352-04-26	105.3, 35.6	130	45	90	7	Huining	[66]
1561-08-04	106.1, 37.5	170	70	164	7^1^/_4_	Wuzhong	[67]
1622-10-25	106.3, 36.5	180	60	90	7	Guyuan	[66,67]
1654-07-21	105.5, 34.3	58	70	20	8	Lixian	[67,68]
1709-10-14	105.3, 37.4	115	50	30	7^1^/_2_	Zhongwei	[66,67]
1718-06-19	105.1, 35.0	150	65	0	7^1^/_2_	Tongwei	[67]
1920-12-16 ^1^	104.9, 36.7	115	90	0	8^1^/_2_	Haiyuan	[59,66,69,70,71]
1920-12-25 ^1^	105.2, 36.6	120	90	18.5	7	Haiyuan	[59,72]
2013-07-22	104.23, 34.52	300	66	47.7	6.6	Minxian–Zhangxian	[73]
2023-12-18	102.81, 35.70	164	46	22	6.3	Linxia	GCMT ^2^

^1^ Focal mechanism was obtained by fitting geometric data of seismic faults, tectonic stress orientations, and rupture scales based on empirical formulae [74]; this method has been used for estimating historical earthquake mechanisms in North China [75]. Geometric information on seismic faults, tectonic stress, and rupture scales, referenced from previous studies, are listed in the last column. ^2^ website: https://www.globalcmt.org/CMTsearch.html (accessed on 25 September 2023).

Besides this, several earthquakes with a magnitude above 5.0 have happened in this region since 1970; according to the catalog compiled by the Department of Monitoring and Forecasting of the China Earthquake Administration, the largest one was the Minxian–Zhangxian *M*6.6 earthquake that occurred on 22 July 2013. According to a comprehensive analysis of the interseismic risk of major active faults in the Chinese mainland by rupture-empty segments, locking degrees, and coulomb stress, several active faults in the NE Tibetan Plateau are considered to have a relatively high seismic risk [76]. Therefore, the spatial *b* value along the faults deserves to be investigated, since the *b* value is an indicator of the underground stress state [3,4,5,6,7].

## 2. Data and Method

### 2.1. Data

The earthquake catalog used for *b* value evaluation was provided by the Gansu, Ningxia, and Shaanxi Earthquake Agency from 1970 to 2008 and the China Seismic Uniform Cataloguing Network from 2009 to June 2023, covering 102.5° E–107.4° E, 34° N–37.7° N and containing the Gansu, Ningxia and Shaanxi provinces, as shown in Figure 1a. Analog observations were used from the 1970s to the 1980s. Since the 1980s, digital seismographs have been deployed, and the China Digital Seismographic Network (CDSN) has been set up [57]. From 1996 to 2000, more stations were set up in the central region, and the monitoring level was improved [77]. Earthquakes above a magnitude of 1.0 in the NE Tibetan Plateau from January 1970 to June 2023 are shown in Figure 1a, and the magnitude–temporal distribution is given in Figure 1b.

The complete magnitude of an earthquake catalog (*M*_C_) is defined as the magnitude of the weakest event that can be fully detected in an earthquake catalog [78]. Since the earthquake frequency is log-scaled with *M*_C_ in the G–R relation, a slight variation in *M*_C_ can significantly change *N*, the number of events above *M*c. If *M*c is overestimated, the number of events used for *b*- value calculation would be greatly reduced, and the accuracy of subsequent computations will be influenced, mainly since the raw catalog is limited. On the other hand, if *M*_C_ is underestimated, the events will be incomplete in low magnitude, making the *b* value result biased.

Several methods have been proposed for *M*_C_ estimation, such as the entire magnitude range (EMR) method [36,79], the maximum curvature (MAXC) method [78,80], the goodness-of-fit-test (GFT) method [80], the *M*_C_-by-*b*-value-stability (MBS) method, and so on. The EMR method performs better than MAXC, GFT and MBS when applied to synthetic test cases or actual data from regional and global earthquake catalogs [79]. However, EMR is very time consuming and is the most computationally intensive. In this research, the Zmap software 6.0 was used to estimate *M*_C_ and its uncertainties [81]. The study region was divided into 0.1° × 0.1° grids, and 1000 times of bootstrap replications were conducted to evaluate the standard deviation.

The research radius and the number of events of each grid are defined according to the amount and spatial distribution of seismic events. A small research radius would lead to a small earthquake sample for each grid, which may lead to the instability of *M*_C_. Conversely, a large earthquake sample needs a big research radius, which can increase the probability of overlapping events along the grid boundaries and can decrease the spatial resolution. After a series of trials with different combinations of the two parameters and making a tradeoff between stability and spatial resolution, each grid’s research radius and the number of events above the minimum magnitude (*M*_C_) were fixed as 50 and 60 km, respectively. The grids with available *M*c results cover most areas of the NE Tibetan Plateau, with the minimum *M*_C_ = 2.6 and the maximum standard deviation = 0.5 (Figure 2a). The number of events was over 100 for each grid, with a magnitude above 2.9 in most fault zones (Figure 2b). The cumulative and non-cumulative number of events above a certain magnitude are also shown in Figure 2c. The *M*_C_ value is estimated as 2.2. The value of *M*c would be raised with time as the monitoring ability is promoted. The time variation of *M*_C_ was also studied by Zmap. Five hundred times of bootstrap replications were conducted for an estimation of the standard deviation of *M*_C_ for each window. From 1970 to 1997, *M*c was about 2.2, with an average standard deviation of 0.2. *M*c has been significantly minimized from 1997 to now (Figure 2d). The results show that *M*_C_ varies not only spatially but also temporally. Another method, named the goodness-of-fit test with a 90-percent possibility (GFT90), was also used to check the distribution of *M*_C_ [80]. The results suggest that a magnitude above 2.4 is complete for most grids in the study region. Based on the research of the completeness of the mainland China earthquake catalog since 1970, the median *M*c was valued at 2.6 from 1 January 1970 to 30 September 2001 [82] and significantly decreased after. To ensure that the catalog used for *b* value calculation is complete, the *M*_C_ used in practice should be slightly larger than the calculated value. Here, we added 0.2 to the maximum *M*_C_ magnitude of 2.6, following Herrmann [83] and as suggested by Woessner and Wiemer [79]. Considering the standard deviation of *M*c, we used *M*_C_ = 2.8, 2.9, 3.0, 3.1 and 3.2 to test the reliability of the *b* value, respectively.

### 2.2. b-Value-Estimation Method

#### 2.2.1. HIST-PPM

Gutenberg and Richter presented the magnitude–frequency relationship, commonly known as the G-R law, in the following manner:(1)$$lgN=a-bM\;(M\geq M_{C}),$$

In this equation, *N* represents the total number of events that satisfy the condition *M* ≥ *M*_C_, while *a* and *b* have constant values. Based on the G–R law, the equation above can be rewritten as follows:(2)$$N\left(M\right)=10^{a-b(M-M_{C})}=Ae^{-\beta (M-M_{C})},$$
in which β = *b*ln10. The following can be used to generate the magnitude probability density distribution:(3)$$f\left(M\right)=\frac{N(M)}{\int_{M_{C}}^{\infty }N\left(M\right)dM}=\beta e^{-\beta (M-M_{C})},$$

The likelihood function for a series of earthquake occurrences with independent magnitudes (*M*1, *M*2, …, *M*n) is as follows:(4)$$L\left(\beta \right)=\prod_{i=1}^{n}f_{\beta }(M_{i})=\prod_{i=1}^{n}\beta e^{-\beta (M_{i}-M_{C})},$$

Ogata proposed the hierarchical space–time point–process model (HIST-PPM), as the *b* value depended on place and time [34,35,84]. This study makes the following assumptions about how *b* relates to the epicenters ($x_{i},y_{i})$:(5)$$\beta =\beta (x_{i},\;y_{i}),$$

Considering that the *b* value is positive, the parametrization of the function $\beta (x_{i},\;y_{i})$ was performed as follows:(6)$$\beta \left(x,y\right)=e^{\emptyset_{\theta }(x,y)},$$

The 2D B-spline function, denoted as $\emptyset_{\theta }$, is defined in reference [35], where θ represents the coefficient of the function $\emptyset_{\theta }$. In this manner, the variable *b* is denoted by a versatile function of position [34,35,36,37,38,85].

In the work conducted by Ogata in the field of HIST-PPM, the study space was divided into tessellations using the Delaunay triangle method, with the triangles being centered at the epicenters of seismic events. The parameter θ was then estimated by maximizing the penalized log likelihood in the following manner [35,84]:(7)R(θ|ω)=lnL(θ)−Q(θ|ω),

The $Q\left(\theta |\omega \right)$ is the penalty term, defined as follows:(8)$$Q\left(\theta |\omega \right)=w\iint \left\{{\left(\frac{\partial \emptyset_{\theta }(x,y)}{\partial x}\right)}^{2}+{\left(\frac{\partial \emptyset_{\theta }(x,y)}{\partial y}\right)}^{2}\right\}dxdy,$$

The weight, denoted as w, is the variable subject to optimization using Akaike’s Bayesian Information Criterion (ABIC) [86,87]. Akaike [87] formulated Good’s technique and established the definition of the Akaike Bayesian Information Criterion (ABIC) based on the notion of entropy maximization [86,87,88].
(9)$$ABIC=-2\max\left(logL\right)+2(number\;of\;hyperparameters)$$

The hypo parameters that yield a reduced Akaike Bayesian Information Criterion (ABIC) value indicate a more optimal fit to the data [35]. Additional information regarding the model-fitting process can be found in the manual for HIST-PPM [84].

Once the *b* values have been acquired at the mesh points, the values within each triangle can be calculated by linear interpolation, utilizing the *b* values present at the triangle’s vertices. This work obtained the spatial *b* value using a resolution of 0.05° × 0.05°.

#### 2.2.2. Zmap

Zmap, developed by Wiemer [81], allows users to examine an earthquake catalog from different perspectives involving *b* values. To evaluate the *b* value with the HIST-PPM method, we also used the Zmap software to explore the *b* value spatial patterns using the maximum likelihood method [89,90]. By comparing to the least square regression, the maximum likelihood technique produces a more reliable estimate [91]. A fixed *M*c and a dynamic *M*c were set up to check the *b* value robustness. As with the *b* value calculation using HIST-PPM, we still set *M*c from 2.8 to 3.2 in 0.1 intervals for the *b* value calculation using zmap. Considering the heterogeneous spatial distribution and uneven distribution of earthquakes in the study area, the number of events for each grid was fixed as 60.

## 3. Results

### 3.1. Spatial b Value of HIST-PPM

The spatial distributions of the *b* value are stable and consistent based on these five different *M*_C_s (Figure 3). Except for minor variations at the boundaries, the low-*b*-value and high-*b*-value regions are basically the same. The low-*b*-value areas are concentrated in the northeast and southwest of the study area near the Zhongwei fault, the Haiyuan fault, and the West Qinling fault. Earthquakes of a magnitude of 5.0 or higher since 1970 correspond well to the low-*b*-value regions. The recently occurred *M*6.3 earthquake in December 2023 was also located in the slightly low-*b*-value area (Figure 3). The high *b* values are shown in Tianzhu City in the northwestern part of the study area and Huating City in the southeastern part (Figure 3).

It was necessary to check whether earthquakes above or equal to a magnitude of 5.0 are prone to happen in the low-*b*-value areas or if the low *b* value is caused by them. In Wang et al. [23], the calculation of spatial *b* values did not include the predicted moderate–large earthquakes occurring in the future. It was found that future moderate–large earthquakes are prone to happen in the low-*b*-value areas. To further clarify this point, we deleted the events above or equal to 5.0 and 5.5 from our catalog and recalculated the *b* value spatial distribution, respectively. It was found that the relatively high- and low-*b*-value areas are quite similar (Figure 4). This means that low-*b*-value areas still match well with moderate–large earthquakes even after they are removed in space, indicating that the occurrences of large earthquakes may not be responsible for lower *b* values, but moderate–large earthquakes are prone to happen in low-*b*-value areas.

When *M*_C_ is selected as 2.8 and 2.9, the *b* value spatial distribution is nearly the same, except for minor details, which proves the catalog is generally complete above these two magnitudes. When *M*c rises to 3.0 and further 3.1 and 3.2, the *b* value spatial pattern changes gradually. High-*b*-value regions in Huating City and around Tianzhu City are narrowed with the decline of *M*_C_, respectively. The main reason for these variances is the loss of samples when elevating the value of *M*_C_. To make a tradeoff between the amount and completeness of the sample catalog, the *b* value based on *M*c = 2.9 was selected as the optimal solution for interpretation.

### 3.2. Spatial b Value of Zmap

The *b* value was also estimated by the maximum likelihood method with a fixed *M*c (from 2.8 to 3.2 with 0.1 intervals) and a dynamic *M*c by Zmap (Figure 5). When *M*c is fixed, each grid has the same *M*c value for *b* value estimation. While using a dynamic *M*c, the *M*c value will be evaluated for each grid. The spatial pattern of the *b* value calculated by Zmap is nearly the same as HIST-PPM, except for some grids with null results due to the fixed sample number of each grid. It should be noted that there is the relatively low reliability of *b* values for regions with a small number of events, as shown in Figure 2b.

High *b* values are revealed at the cities of Huating and Tianzhu, which is also shown in the HIST-PPM results (Figure 3). Quarry blasts and coal-mining activity are significant in these two regions [92,93,94]. Gulia and Gasperini [95] applied the D/N method to analyze the *b* value in mining regions, as suggested by Wiemer and Baer [96]. To investigate the high *b* values in these two regions further, we used the D/N method as well. In the Huating region, D/N is 0.97 (Figure 6). The Huating region is known for its coal-mining activity. To complete the whole coal-mining procedure, the mining operates around the clock for three shifts (Goldthorpe, 1959) [97]. The mining seismicity at nighttime is comparable to daytime, and D/N is not like that of the blast region. In the Tianzhu region, D/N is 3.72 (Figure 6). This region is known for limestone mining [98], and a much higher seismicity in the daytime is shown. This is typical for quarry-blast areas [96]. The significant difference in D/N between these two regions is most likely due to their working hours.

It is widely known that the depth of mining seismicity is shallow. To further check the high *b* value in these regions, we removed the mining seismicity by deleting the events less than and equal to 3 km from the catalog of these two regions. After 30% and 35% of the events were removed from the catalog in the Huating and Tianzhu regions, respectively, the *b* value was estimated by HIST-PPM again. A high *b* value still exists in these two regions (Figure 7), and no strong earthquakes have happened in these two regions since 1970.

The results of HIST-PPM and Zmap are consistent in most areas; however, HIST-PPM displays more details. As the Zmap results are based on grid searches, samples associated with various fault systems may be involved to match the grid search criteria, which causes a bias of the *b* value, especially for highly seismic-clustered regions with different types of active faults. HIST-PPM may be more objective and accurate in computing spatial *b* values in such regions, since it uses Delaunay tessellation to interpolate the value at the nearest three earthquakes and does not call for the gridding of seismic events [23,34,84]. Consequently, considering both the advantages of the methodology and the tradeoff between the amount and completeness of the sample catalog, the HIST-PPM result of *M*c = 2.9 was adopted as the optimal resolution for further interpretation.

## 4. Discussion

Moderate–strong earthquakes above 5.0 from 1970 to 2023 are plotted in Figure 3 and Figure 4. The strongest earthquakes are located in or at the boundary of the low-*b*-value areas and are close to the main faults, implying that the *b* value may reflect the state of underground stress and probably provides forecast information for long-term earthquakes [8,9].

A series of NWW fault belts allot in the NE Tibetan Plateau, such as the Zhongwei fault (sometimes called the Xiangshan–Tianjingshan fault belt), the Haiyuan fault belt, and the Western Qinling fault belt, accommodating the eastward movement of Tibet relative to the Gobi–Alsa Shan platform to the north (Figure 1). The *b* value varies systematically for different faulting styles and is controlled by different stress regimes [11,12]. Therefore, we divided the NE Tibetan Plateau into four regions according to the strike and the nature of fault belts. The first region is adjacent to the Zhongwei fault and the Luoshan fault, called ZW-LSFR; the second region is the Haiyuan fault belt and its adjoining area, abbreviated as HYFR. The third one is the area between the Haiyuan fault belts and the Western Qinling fault belts, with a few mid-scale faults near Lanzhou City, called the Longxi fault region (LXFR). The last region is the Western Qinling fault region, abbreviated as WQLFR.

### 4.1. ZW-LSFR

The Zhongwei fault (ZWF) starts from Gulang and extends southeastward and merges with the Luoshan fault, extending for 240 km [99]. ZWF is an oblique left-lateral fault with a significant reversal component. Its strike-slip decreases eastward from 3.5 to 0.5 mm/yr, which is accomplished by an increase in the vertical rate from 0.5 to 0.8 [100,101,102,103]. LSF runs along the eastern side of the Luo Mountains [104] and was thought to be a left-lateral strike-slip fault in early research [105,106], but numerous pieces of evidence of right-lateral offsets have been discovered [104,107,108]. Our results show a gradual decrease in the *b* value eastward, consistent with the increase in the reversal component of ZWF, since the *b* value is generally lower in the compressional regions [11]. The low *b* value clamped between the LSF and ZWF is significant (Figure 8). The decreases in the strike-slip rate of ZWF and the GPS velocity rate both suggest that this region may be under a high-stress state and coincides with low *b* values [100]. Three earthquakes above a magnitude of 5.0 have happened in this low-*b-*value region since 1970.

### 4.2. HYFR

The Haiyuan fault zone consists of nine subparallel fault strands separated by pull-apart basins of different sizes [55,109]. According to the geometry pattern and geomorphology, the Haiyuan fault can be divided into three segments and is predominantly of left-lateral strike [110] (Figure 9). The slip of the Haiyuan fault varies systematically, too. For instance, the quaternary slip rate estimates are 12 ± 4 mm/yr, 8 ± 2 mm/yr, and ~5 mm/yr or less [50,101,111,112] on the central and eastern segments of the Haiyuan fault. However, the modern slip- or strain-accumulation rate derived from GPS and interferometric synthetic aperture radars suggests a low rate of 5~8 mm/yr [54,113]. The rupture length of the 1920 *M*8^1^/_2_ Haiyuan earthquake is about 230 km [55,56]. A slightly higher *b* value than surrounding areas along the rupture zone is apparent and indicates the highly fractured zone caused by this larger earthquake and the following 1920 *M*7 earthquake nearby (Figure 9). The variations of the *b* values are apparent, which corresponds to the segmented pattern of the Haiyuan fault. The western segment of the Haiyuan fault consists of five fault strands, and the eastern consists of three. The middle segment consists of only one fault strand, which makes it different from the western and eastern segments. These geometric segments may cause a heterogeneity of *b* values along the Haiyuan fault. The west end, at Jingtai City, and the east end, at Guyuan City, of the Haiyuan fault have low *b* values and are far from the epicenter. A 3D curved-grid finite-difference method was used to inverse the dynamic rupture process of the 1920 *M*8^1^/_2_ Haiyuan earthquake [114]. The region with the maximum horizontal peak ground velocity and the highest intensity shows a small high-*b*-value area near Haiyuan, which indicates that stress is ultimately released during this large earthquake. Moreover, the locations of six earthquakes above a magnitude of 5.0 coincide with low-*b*-value areas (Figure 9).

### 4.3. LXFR

Three fault strands suit this region: the ZLHF fault in the north, the MXSF fault in the south, and the Huining fault near Huining City in the east [115]. MXSF is a Holocene-active fault with a left-lateral strike and reverse slip. Significantly low *b* values are observed along the MXSF fault, and one earthquake with a magnitude of 5.0 has happened since 1970 (Figure 10). GPS data show a gradual increase in the reverse slip from the north to the middle strand of MXSF [64]; this may have caused the lower *b* value along the middle part of MXSF. While joining with ZLHF northward, it changed into a higher *b* value, and a magnitudeof 6.1 earthquake happened at the high- and low-*b-*value transition zones. Eastward to the Huining fault near Xiji City, an apparently low *b* value is shown within a small region. Another earthquake with a magnitude of 5.5 took place precisely in this region. The *b* value spatial distribution still displays a consistent pattern with notable earthquakes and fault kinematic characteristics. However, the low number of events in this region should be noticed (Figure 2b). Although a more current *b* value can be achieved even in a low number of event areas based on the HIST-PPM algorithm [35], the uncertainties of these grids cannot be neglected. Therefore, more detailed research needs to be conducted on the physical interpretation of *b* value variations in this region.

### 4.4. WQLFR

The NWW-trending West Qinling fault, the main fault in this region, is one of the primary active strike-slip faults developed in the NE Tibetan Plateau [116,117]. According to the historic earthquake catalog, several large earthquakes have occurred along the West Qinling fault, such as the 143 *M*7¼ Western Gangu earthquake and the 734 *M*7½ Tianshui earthquake (Figure 1). GPS data show that the West Qinling fault belt is fully stuck with a high seismic risk [118], and a significantly low *b* value along WQLF is observed in Figure 11, both of which make it easy to understand that the West Qinling fault region is under a low-stress state. Five earthquakes above a magnitude of 5.0, including the 2013 Mingxian–Zhangxian *M*6.6 earthquake, are all in this low-*b*-value region.

## 5. Conclusions

In this study, after obtaining the completeness magnitude of the regional earthquake catalog in the NE Tibetan Plateau since 1970, long-term spatial *b* values of the NE Tibetan Plateau were estimated using HIST-PPM and ZMAP, respectively. To test the influence of *M*c and the robustness of the *b* values, we deployed *M*c = 2.8, 2.9, 3.0, 3.1 and 3.2 during the estimation of the *b* values, respectively. Except for some minor unconformities caused by a decrease in samples, the low-*b*-value areas are well consistent with each other (Figure 3 and Figure 5). And the spatial *b* value distribution of HIST-PPM agrees with that of Zmap in general.

The results indicate a high spatial heterogeneity in *b* values within the study area. The northwestern and southeastern regions exhibit higher *b* values. Along the Haiyuan fault zone, the central rupture zone of the Haiyuan earthquake has relatively higher *b* values than other regions of the fault zone, which is possibly related to the sufficient release of stress during the main rupture of the Haiyuan earthquake. The *b* values vary from high in the west to low in the east along the Zhongwei fault. On the West Qinling fault zone, the epicenter of the recent Minxian–Zhangxian earthquake is associated with a low *b* value. In general, regions with low *b*-values correspond well to areas with moderate–strong seismic events in the past 50 years. The spatial differences in *b* values may reflect variances in seismic hazards among fault zones and regions within the same fault zone.

## Figures and Tables

**Figure 2 entropy-26-00182-f002:**
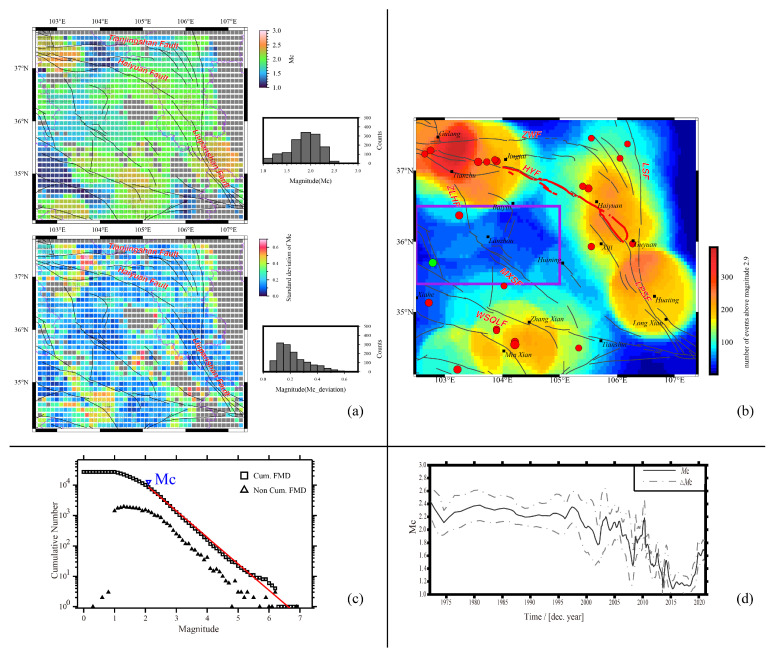
(**a**) The upper map shows the spatial distribution of *M*_C_ of the NE Tibetan Plateau based on the catalog from 1970 to June 2023, and the lower map shows the standard deviation of *M*c. The gray boxes denote the grids with null results; the black dots represent the events shown in Figure 1. The purple and black solid lines are provincial borders and active faults [51]. (**b**) The number of events above magnitude 2.9 for *b* value estimation; purple lines encircle the region with a low number of events. (**c**) The frequency and magnitude distribution of the catalog; the squares represent the accumulated number of events above a certain magnitude, and the triangles represent the non-accumulated number of events. The red line represents the best fitted relationship of log*N* = a − *bM* calculated by Zmap. (**d**) The time variations of *M*c; the solid black line represents *M*c, and the dashed line represents *M*c’s standard deviation.

**Figure 3 entropy-26-00182-f003:**
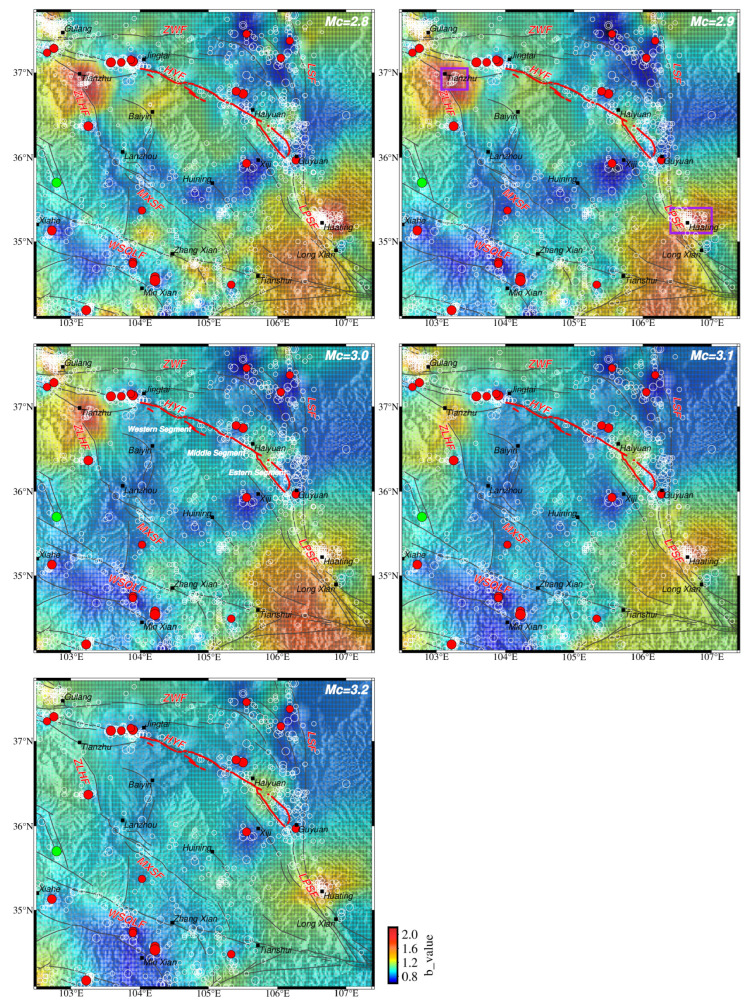
The spatial *b* value distribution is based on different *M*cs by HIST-PPM. The red solid circle denotes earthquakes above and equal to a magnitude of 5.0 since 1970 in the NE Tibetan Plateau. The white hollow circle is the event above the *M*c. Dark gray solid lines are the main fault trace in the NE Tibetan Plateau, modified from Deng [51]; the abbreviated name for each fault trace is the same as Figure 1. Bold red lines denote the 230 km rupture zone of the 1920 *M*8^1^/_2_ Haiyuan earthquake [55,56]. Main cities and counties are also shown. Solid purple lines encircle the events with high *b* values. The green solid star indicates the 18 December 2023 *M*6.3 earthquake in the Linxia Hui autonomous region in Gansu, which was not included in the *b* value estimation.

**Figure 4 entropy-26-00182-f004:**
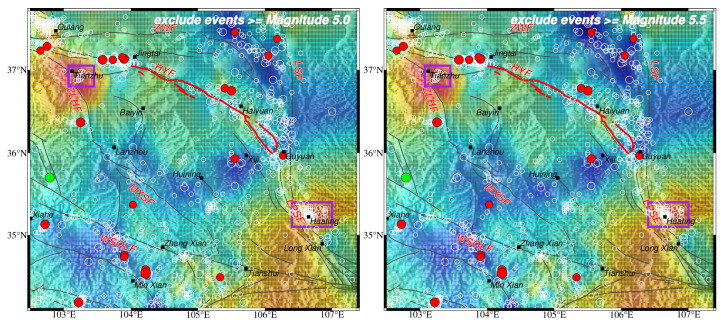
Spatial distribution of *b* value based on events with *M*c ≤ *M* ≤ 5.0 (**left**) and events with *M*c ≤ *M* ≤ 5.5 (**right**).

**Figure 5 entropy-26-00182-f005:**
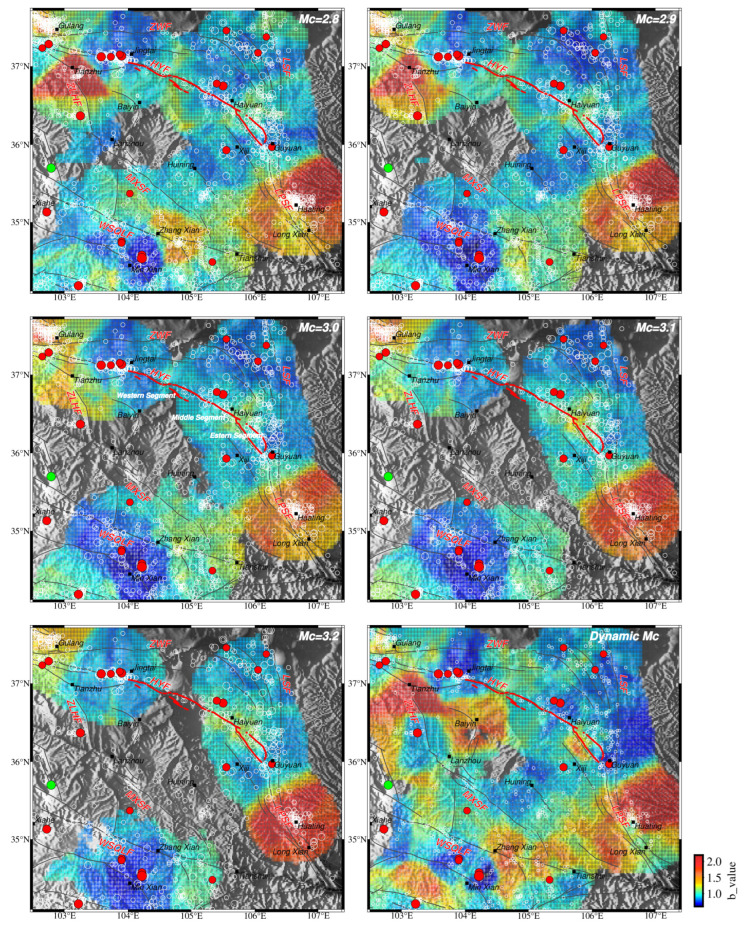
The spatial *b* value distribution is based on different *M*cs by Zmap. Other factors are the same as in Figure 3.

**Figure 6 entropy-26-00182-f006:**
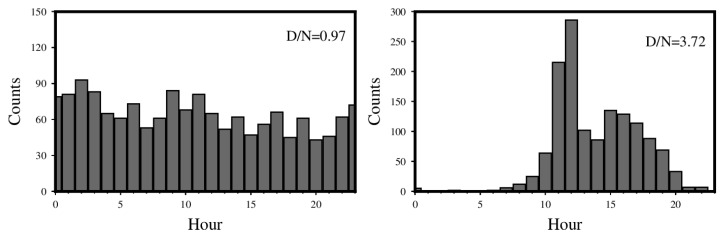
Histograms of the hourly events in the cities of Huating and Tianzhu. These two regions are encircled with purple lines in Figure 7. The value of D/N is shown in the upper right corner for each.

**Figure 7 entropy-26-00182-f007:**
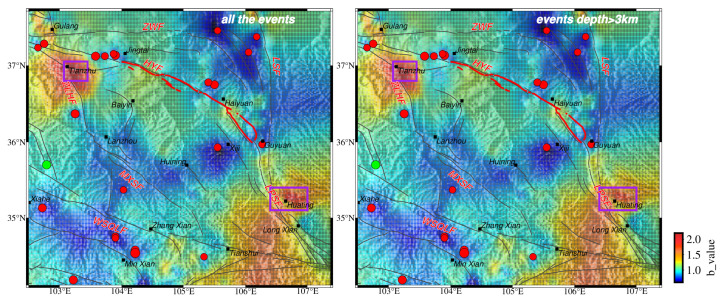
The spatial distribution of the *b* value estimated by all the events (**left**) and by excluding events with depths less than and equal to 3 km in the Huating and Tianzhu regions (**right**). These two regions are encircled with purple lines.

**Figure 8 entropy-26-00182-f008:**
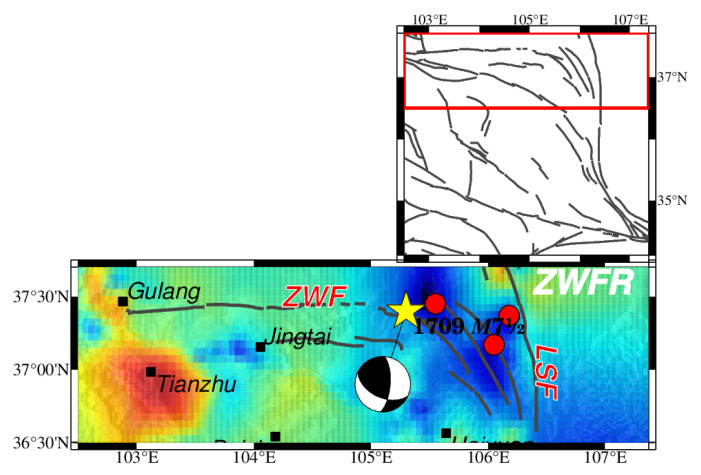
The spatial distribution of HIST-PPM *b* values concentrated around ZWF and LSF. The color bar for the *b* value and other elements is illustrated in Figure 3 and is not repeated here. The red frame in the upper map encricle the ZWFR in large scale. The yellow star represents the historic earthquake, which is also shown in Figure 1.

**Figure 9 entropy-26-00182-f009:**
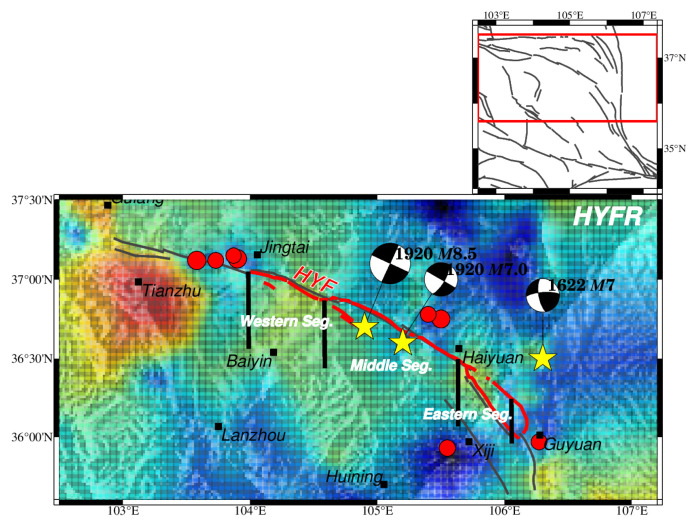
The spatial distribution of HIST-PPM *b* values concentrated around HYF. Bold black lines denote the segmentation of HYF [110]. The yellow stars denote the 1920 *M*8^1^/_2_ Haiyuan earthquake and the 1920 *M*7 Haiyuan earthquake. The color bar for the *b* value and other elements is illustrated in Figure 3 and is not repeated here. The red frame in the upper map encricle the HYFR in large scale. The yellow stars represents the historic earthquakes, which are also shown in Figure 1.

**Figure 10 entropy-26-00182-f010:**
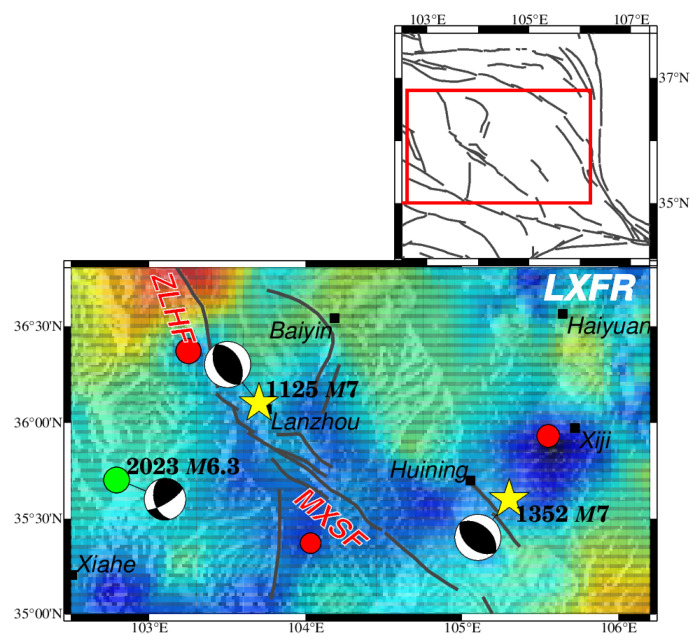
The spatial distribution of HIST-PPM *b* values concentrated around LXFR. The color bar for the *b* value and other elements is illustrated in Figure 3 and is not repeated here. The red frame in the upper map encricle the LXFR in large scale. The yellow stars represents the historic earthquakes, which are also shown in Figure 1.

**Figure 11 entropy-26-00182-f011:**
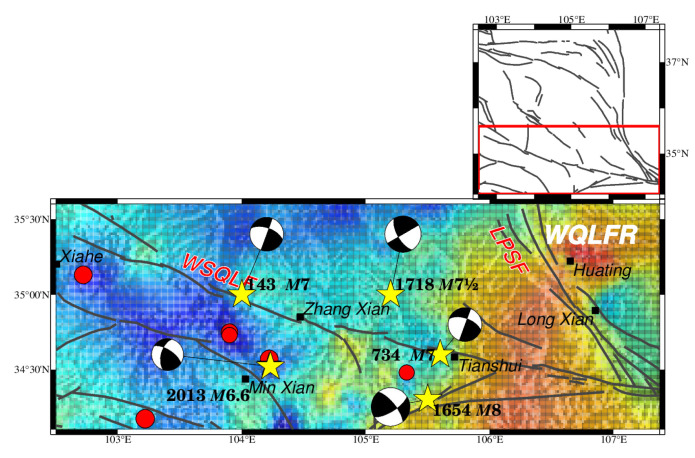
The spatial distribution of HIST-PPM *b* values concentrated around WQLFR. The yellow star represents the 2013 *M*6.6 Minxian–Zhangxian earthquake. The color bar for the *b* value and other elements is illustrated in Figure 3 and is not repeated here. The red frame in the upper map encricle the WQLFR in large scale. The yellow stars represents the historic earthquakes, which are also shown in Figure 1.The high degree of spatial heterogeneity in *b* values can provide clues for observing the segmented features of fault zones, which are usually proved by geological methods. Earthquakes usually cluster along the fault belts, providing sufficient samples for the *b* value and making the *b* value more accurate here. Therefore, it is suggested that regions far away from faults may not acquire reliable information on *b* values for regions including complex fault systems.

## Data Availability

The catalog in this study is provided by the Gansu, Ningxia, and Shaanxi Earthquake Agency from 1970 to 2008 and the China Seismic Uniform Cataloguing Network from 2009 to June 2023.

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
