# Peer review of "Spatial Heterogeneity of Values in Northeastern Tibetan Plateau and Its Interpretation"

_entropy, 2024, doi:10.3390/e26030182_

Round 1
Reviewer 1 Report
Comments and Suggestions for Authors
See attached file

Reviewer 2 Report
Comments and Suggestions for Authors
The paper is interesting and of potential interest for the readers of Entropy. The underlying computational methodology is rather complex and cannot be described in details in the text: this implies that reading original papers by Ogata et al. is mandatory to better understand the paper.
The reference list is extensive but sometimes incomplete: some bilbliographic elements (e.g., the title if the journal) are laking for references 4, 8, 22,23,25,26,31,39,52,53,55, 58, 78, 79, 84,87,113. Anyway outcomes are clearly described and discussed.
My only concern relies on the claimed association between low b values and occurrence of Ml>5.0 earthquakes. In the paper, this association supports the idea that areas prone to future earthquakes are characterized by low b values. However, since the considered Ml>5 earthquakes are included in the catalogue used for computing b values, one could suspect that those occurrences (and respective aftershock sequences) are actually responsible for lower b values. I suggest that this possibility is discussed in the text.
Comments on the Quality of English LanguageI suggest a revision of the english writing. Locally, some statements should be rephrased to avoid misunderstandings
Round 2
Reviewer 1 Report
Comments and Suggestions for Authors
In this second version of the manuscript, the authors performed a lot of new computations and sensitivity analyses to clarify some points raised by myself and the other reviewer. They also changed a little bit their conclusions, and added the following statement "the uncertainties of these grids cannot be neglected. Therefore, more detailed research needs to be done on the physical interpretation of b-value variations in this region." to underline that some spatial b-value variations cannot be interpreted. I appreciate their honesty. Overall, this second version is a great improvement with respect to the first one. All the newly performed computations make the manuscript scientifically sound. Therefore, I suggest accepting the paper.